# A Cretaceous Chafer Beetle (Coleoptera: Scarabaeidae) with Exaggerated Hind Legs—Insight from Comparative Functional Morphology into a Possible Spring Movement

**DOI:** 10.3390/biology12020237

**Published:** 2023-02-02

**Authors:** Yuanyuan Lu, Dirk Ahrens, Chungkun Shih, Josh Jenkins Shaw, Xingke Yang, Dong Ren, Ming Bai

**Affiliations:** 1Key Laboratory of Zoological Systematics and Evolution, Institute of Zoology, Chinese Academy of Sciences, Beijing 100101, China; 2Zoologisches Forschungsmuseum A. Koenig, Adenauerallee 127, 53113 Bonn, Germany; 3College of Life Sciences and Academy for Multidisciplinary Studies, Capital Normal University, Beijing 100048, China; 4Department of Paleobiology, National Museum of Natural History, Smithsonian Institution, Washington, DC 20013-7012, USA; 5Natural History Museum of Denmark, Zoological Museum, Universitetsparken 15, 2100 Copenhagen, Denmark; 6College of Plant Protection, Hebei Agricultural University, Baoding 071001, China; 7School of Agriculture, Ningxia University, Yinchuan 750021, China; 8Northeast Asia Biodiversity Research Center, Northeast Forestry University, Harbin 150040, China; 9University of Chinese Academy of Sciences, Beijing 100049, China

**Keywords:** scarab beetles, exaggerated hind legs, springing, fighting, phytophagous, Cretaceous fossil

## Abstract

**Simple Summary:**

Exaggerated morphological structures are fascinating for evolutionary biologists and the public, and scarab beetles in particular are famous for their diverse exaggerated characters. Here, we report a new genus and species of Mesozoic scarab beetle with unusually robust and structured hind legs and fine color marking patterns on the dorsal and ventral surfaces of the body. Based on morphological characters, we performed phylogenetic and morphometric analyses. The results support the placement of this new taxon in the pleurostict lineage of scarab beetles, consequently representing one of its earliest records. We hypothesize that the exaggerated leg structures supported springing movements and fighting. Furthermore, the unusual marking patterns of this fossil suggest that the new taxon exhibited diurnal foraging behavior, potentially visiting leaves or flowers of Lower Cretaceous plants. This study provides new insights into the exaggerated structures of Mesozoic insects and the timing of the evolution of this diverse beetle family.

**Abstract:**

The phenomenon of exaggerated morphological structures has fascinated people for centuries. Beetles of the family Scarabaeidae show many very diverse exaggerated characters, for example, a variety of horns, enlarged mandibles or elongated antennal lamellae. Here, we report a new Mesozoic scarab, *Antiqusolidus maculatus* gen. *et* sp. n. from the Lower Cretaceous Yixian Formation (~125 Ma), which has unusually robust and structured hind legs with greatly enlarged spurs and a unique elongated apical process. Based on simulations and finite element analyses, the function of these structures is hypothesized to support springing to aid movement and fighting. Based on available morphological characters, we performed phylogenetic analyses (maximum parsimony) of the main subfamilies and families of Scarabaeoidea. The results support the placement of *Antiqusolidus* gen. n. as a sister group of Rutelinae within the phytophagous lineage of pleurostict Scarabaeidae. Furthermore, the unusual delicate color marking patterns in the fossil specimens suggest that the new species might have been diurnal and potentially visited the leaves or flowers of Early Cretaceous plants. This morphological and functional study on this extinct scarab species provides new sights into exaggerated structures in Mesozoic insects.

## 1. Introduction

Exaggerated body structures such as elongated appendages and large outgrowths have always attracted the interest of biologists and the public, especially to seek the extreme boundaries of marvelous nature [1]. As a highly diversified group, insects provide an unlimited number of examples: different kinds of horns in scarab beetles (Dynastinae, Scarabaeinae/Coleoptera) and treehoppers (Membracidae/Hemiptera); tremendously enlarged heads in soldier ants (Formicidae/Hymenoptera) and termites (Isoptera/Blattodea); developed mandibles in stag beetles (Lucanidae/Coleoptera); a transversely expanded body in lace bugs (Tingidae/Hemiptera); or modified predatory forelegs in mantis (Mantodea) [2,3,4,5,6,7,8]. Exaggerated structures have also been recorded in extinct insects, such as the extremely extended abdominal segments in Jurassic Mecopterans [9] or oversized mandibles in unicorn ants from the Cretaceous amber of Myanmar [10]. All of these structures triggered scientists’ search for answers on the how, when and why of their origin and evolution. Moreover, the records of extinct taxa fill knowledge gaps about the development of these characters and the corresponding species/lineages.

The functions of exaggerated structures in insects can be classified into three types: sexual selection, locomotion/predation/feeding and sociality [1]. Among the exaggerated structures, hind legs are normally modified for special locomotion or resource competition [11,12,13]. To date, there are still only a few records of extinct insects with exaggerated or strongly modified metatibial spurs, although they can be found in many extant beetles. However, our knowledge about the function of exaggerated structures of the hind legs based on functional morphology studies is limited to some scattered reports [14].

In this paper, we describe a new scarab beetle from the Lower Cretaceous Yixian Formation (~125 Ma) that has unusually robust and structured hind legs with greatly enlarged spurs and an elongated process [15]. By using finite element analyses (FEA), we investigated the possible function of these exaggerated hind legs. We evaluate two possible hypotheses, “springing” and “fighting”, in order to explain the development of these exaggerated structures. In addition, we also inferred the phylogenetic position based on morphology and discuss the possible biology of the new taxon. We found that this rare scarab beetle belongs to the phytophagous lineage of pleurostict Scarabs (Coleoptera: Scarabaeidae) [16], and the distinct color marking patterns suggest that the species ha d diurnal habits, possibly visiting the exposed parts of Early Cretaceous plants. This new taxon enriches our knowledge of exaggerated characters, the movement habits of beetles and the functional biodiversity of insects in the past.

## 2. Materials and Methods

### 2.1. Material

This study is based on a new fossil taxon, *Antiqusolidus maculatus* gen. *et* sp. n. Two specimens were collected from the Yixian Formation: the holotype was collected near Liutiaogou Village, Ningcheng County, Chifeng City, Inner Mongolia, China; the paratype was collected near Dawangzhangzi village, Lingyuan City, Liaoning Province, China. The age of the Yixian Formation is regarded as the Early Cretaceous (latest Barremian to earliest Aptian) and linked to radioisotope dating of ca. 125 Ma [15,17,18,19,20,21]. The holotype and paratype specimens are deposited in the Key Lab of Insect Evolution & Environmental Changes, College of Life Sciences, Capital Normal University (CNUB, Curator Dong Ren), Beijing, China.

### 2.2. Nomenclatural Acts

This published work and the nomenclatural acts it contains have been registered in ZooBank, the online registration system for the International Code of Zoological Nomenclature (ICZN). The LSIDs for this publication is: urn:lsid:zoobank.org:pub:EB64325E-1691-46B3-9DBE-49452152317C; urn:lsid:zoobank.org:act:EF991CAE-B068-4C3C-A393-6A77FBD72834; urn:lsid:zoobank.org:act:85F50205-DD91-4AFB-B92F-5553F1027D2C.

### 2.3. Images

Observations were carried out under an Olympus SZ61 stereomicroscope. Digital images were created with a Canon 5D digital camera in conjunction with a Canon MP-E 65 mm f/2.8 1-5X Macro Lens fitted to a macro rail (Cognisys).

### 2.4. Phylogenetic Position

The character matrix is based on previously published studies [22,23,24]. Thirty-seven (37) taxa of Scarabaeoidea were selected, including two outgroup taxa: *Sternolophus rufipes* (Hydrophilidae) and *Hister* sp. (Histeridae). A total of 34 extant species of all eight families of Scarabaeoidea, including six subfamilies of Scarabaeidae, are included, as well as *Antiqusolidus maculatus* gen. *et* sp. n. The data matrix (53 characters × 37 taxa) was used to conduct a phylogenetic analysis (Appendix A) using maximum parsimony [25].

Tree search with maximum parsimony (MP) analysis was run with equal weighting (EW) in TNT (version 1.5) [26] using the following settings [27]: Analyze > “traditional search”; “max. Tree” = 500,000; “random seed” = 1000; “number of additional sequences” = 50,000; “trees to save per replication” = 10. Additionally, “tree bisection reconnection (TBR) was utilized as the permutation algorithm of the branches. All characters were treated as unordered and nonadditive. In our analyses, the outgroup taxon *Sternolophus rufipes* was chosen to root the tree. The strict consensus tree was calculated based on resulting MPT trees.

### 2.5. FEA Simulation for the Bumping Protection Function

#### 2.5.1. Testing Hypotheses

A finite element analysis (FEA) was applied to evaluate the function of the exaggerated spurs in *Antiqusolidus maculatus* gen. *et* sp. n. The first possible hypothesis of the “springing” function was tested (for details, see Discussion section) based on a comparison with an extant beetle, *Aphthonoides armipes* Bryant, 1939 (Alticinae), and two non-springing species (*Repsimus manicatus* (Swartz, 1817); *Gastroserica kucerai* Ahrens, 2000). The first also has exaggerated spurs that are significantly stronger than normal-sized spurs and is reported to spring. The hind tibiae and enlarged spurs are the structures involved in the bumping process of the *Aphthonoides* species. Three-dimensional models of the hind tibia and spurs were compared using FEA.

#### 2.5.2. The 3D Model Reconstruction of Legs

Since the new fossil taxon is preserved as compression fossils, the 3D morphology of the fossil specimen is not well preserved. Therefore, it is not possible to infer its functional morphology directly. Therefore, we first created a 3D model based on the fossil specimen. Secondly, we generated three other 3D models for comparison based on a selection of extant specimens (see above) of Alticinae (Chrysomelidae) and Melolonthinae/Rutelinae (Scarabaeidae). *Aphthonoides armipes* has a very long spur on the hind tibia, very similar to the fossil species. *Repsimus manicatus* (Rutelinae) has exaggerated hind legs only in males. *Gastroserica kucerai* (Melolonthinae) has spurs of normal size and is not known to exhibit jumping or fighting behavior. The specimens were scanned with a MicroXCT 400 (Carl Zeiss X-ray Microscopy Inc., Pleasanton, CA, USA) at the Institute of Zoology, Chinese Academy of Sciences. Scans of the hind legs were performed with a beam energy of 60 kV, absorption contrast and spatial resolutions of 1.5625 μm, 29.5858 μm and 10.1215 μm. All 3D models were created using visage Imaging Amira 5.2 and Geomagic Studio 12. Segmentation was performed using a combination of automatic thresholding based on gray-scale values and manual corrections in the three orthogonal views.

#### 2.5.3. FEA Simulation

A triangulated surface mesh was created and smoothed for each 3D model. The surface files were imported into Z88Aurora V3 (http://www.z88.de/ (accessed on 15 January 2020)) for the FEA. A tetrahedral volume mesh was produced using the settings of NETGEN, Tetrahedrons (linear) and value 5. Force loads were applied to the hind tibial base, and constraints were set at the tips of the spursfor elastic, linear and homogeneous materials. The setting of SORCG with von Mises stress failure theory was selected in the solver step [28]. The figures of the stresses at corner nodes were saved in the postprocessing step. The material properties followed Sun et al. [29], with a Young’s modulus of 3.74 ± 0.73 GPa. A Poisson ratio of 0.3 was used here, which was measured for the lobster cuticle [30]. The density of the cuticle of this species is unknown. We chose the density value (0.89 × 10^3^ kg/m^3^) of *Cybister* (Coleoptera: Dytiscidae) for this study [31,32].

In the FEA part, the result of von Mises stress must be identical if one changes the values of material properties using elastic, linear and homogeneous material settings. However, the material properties only affect displacements and strains, whereas von Mises stresses are only related to the external forces being applied [33]. As we are only interested in comparing the difference in Mises stresses among the models, which could reflect the mechanical difference between the structures, the results will be sound and comparable among all models if we apply the same force on the same boundary conditions and the same material properties. Herein, we chose 5N as the force for all models in accordance with Goyens et al. [34], which represents the bite force of the mandibles in Lucanidae (Coleoptera).

### 2.6. Morphometric Analyses for the Fighting Function of the Hind Legs

To investigate the hypothesis that the enlarged hind legs serve the “fighting” function, two morphometric indices previously used for the investigation of such a hypothesis in monkey beetles [35] were used in this study to test the degree to which possible fighting may be linked with the exaggerated hind leg structures. The femur shape index (FSI, width/length) and tibial shape index (TSI, width/length) are defined by the shape of the femur and tibia [35]. The indices calculated for the four studied species (*Aphthonoides armipes*, *Antiqusolidus maculatus* sp. n., *Gastroserica kucerai* and *Repsimus manicatus*) were compared with data from 1512 male monkey beetle specimens of 37 species (Scarabaeidae: Hopliini) taken from the literature, for which sexual dimorphism in relation to leg weaponry or not (fighting Hopliini and non-fighting Hopliini) was exhaustively studied (Appendix A) [35,36]. Indices were also mapped into box plots.

## 3. Results

### 3.1. Systematic Paleontology


Coleoptera Linnaeus, 1758Family: Scarabaeidae Latreille, 1802Subfamily: *incertae sedis*



**Genus: *Antiqusolidus* Lu, Ahrens, Bai, Shih & Ren gen. n.**


LSID: urn:lsid:zoobank.org:act:EF991CAE-B068-4C3C-A393-6A77FBD72834

*Type species*. *Antiqusolidus maculatus* Lu, Ahrens, Bai, Shih & Ren sp. n.

*Etymology*. From Latin words “antiqu-” and “solidus”, meaning ancient and strong scarab beetle. Gender is masculine.

*Diagnosis.* Large-sized body, nearly elliptical; frontoclypeal suture distinctly impressed; antenna with a club; outer edge of protibia with three teeth; metacoxal plates large, at least as long as metafemur is wide; metafemur much wider than mesofemur; mesotibia extremely short, less than half length of mesotarsus, preapical surfaces of mesotibia with transverse or oblique ridges or combs; first segment of mesotarsus longest; metatibia very short too, much shorter than metafemur, with two exceptionally strong apical spurs, spurs nearly equal in length, tightly close; side of metatibia with a long process; metatarsus longer than metatibia, first tarsomere longer than any of the others; abdominal process acute or narrowly rounded.


***Antiqusolidus maculatus* Lu, Ahrens, Bai, Shih & Ren sp. n. (Figure 1 and Figure 2)**


LSID: urn:lsid:zoobank.org:act:85F50205-DD91-4AFB-B92F-5553F1027D2C

*Etymology.* The specific name is derived from the Latin word “*maculatus*-”, which means markings, referring to its unique pattern of markings.

*Material.* Holotype, CNU-COL-NN2011001p/c (part and counterpart). Paratype, CNU-COL-NN2011002. Both from the Yixian Formation: holotype collected in Inner Mongolia, China; paratype collected in Liaoning Province, China (for details, see Section 2.1 Material).

*Diagnosis.* Same as the genus.

Description. Holotype. Body shape elongate ovoid. Length: 22.4 mm (from the apex of the clypeus to the apex of the elytra); maximum width at middle of elytra: 12.4 mm; pronotum narrower than elytra; with many markings and color patterns of stripes and large spots on the pronotum, elytra, pygidium, abdomen and legs (Figure 1a–d).

Head. Length: 3.5 mm; width: 5.0 mm; narrower than pronotum; clypeus sub-trapezoid, front nearly straight, with broad, rounded angles; frontoclypeal suture curved; eyes large; mandible protruding beyond labrum (Figure 1g); labrum and clypeus well separated, with their principal surfaces both facing dorsally.

Pronotum. Length 4.8 mm, width 8.8 mm, 1.8 times as wide as long, widest part at the middle, slightly narrower than elytra, obviously wider than the head; anterior emarginated, front angles acute and slightly protruding, hind angles obtuse; three distinctly large symmetrical spots on either side of the midline, two patches at the midline and arranged up and down (Figure 1h).

Scutellum. Triangular, posterior apex acuminate.

Elytra. Length 14.4 mm, width 6.2 mm, about 2.3 times as long as wide, widest at the middle; with nine visible longitudinal striae formed by regular, symmetrical large patches (between first stria and fourth stria; between second stria and sixth stria; between sixth stria and seventh stria; on ninth stria) (Figure 1a,c).

Pygidium. Length 2.7 mm, width at base, 5.6 mm; exposed and transverse, slightly convex; with two large triangular markings, meeting at the middle.

Ventral thoracic surface covered with three large triangular markings (Figure 1b,d).

Abdomen. Length 8.2 mm; abdominal process acute and narrowly rounded; with six visible ventrites and 2–6 with banded markings at base; first ventrite length 2.2 mm, second 0.8 mm, ventrites of both third and fourth 0.9 mm, fifth 1.4 mm, and the last 1.5 mm.

Legs. Robust and strong, especially hind legs. Profemur about 2 times as wide as long; protibia with three outer teeth and one inner spur; protarsus preserved with only three segments in this specimen (Figure 1e). Mesofemur 2.3 times as wide as long, with a large patch; mesotibia extremely short, less than half of mesotarsus; mesotarsus with five segments, first segment longest (Figure 1i). Hind legs enlarged; metafemur 2 times as wide as long, with a large oval patch, almost near margin; the outer surface of metatibia slightly concave, inner terminal extending to the tarsal median and forming an elongated process, end of inner side with one irregular spot; metatibia with two exceptional strong apical spurs, much stronger than spurs in mesotibia, longer than half length of metatarsi, two spurs nearly equal in length, tightly close, almost equal to the elongated process; metatarsus longer than metatibia, with five segments, the first one longer than all the others (Figure 1f,j).

Measurements. Body length/body width/head length/head width/pronotum length/pronotum width/elytra length/elytra width, in mm: CNU-COL-NN2011001: 22.4/12.4/3.5/5.0/4.8/8.8/14.4/6.2.

*Morphological variability*. In the paratype, the basic pattern of markings in pronotum and elytra are the same but variable in detail (Figure 2a–c). The elongated process is located on the outer side of the metatibia and is shorter than the holotype specimen, about half of the length of spurs. Body length/body width/head length/head width/pronotum length/pronotum width/elytra length/elytra width, in mm: CNU-COL-NN2011002: 21.6/11.8/3.2/4.8/4.3/8.7/14.2/5.7, a little bit smaller than the holotype.

*Notes*. The sexes of the holotype and paratype are unknown due to limited preservation. From the characters of the hind legs, it is highly possible that the holotype and paratype are of opposite sexes (see Discussion section).

### 3.2. Comparative Morphology and Phylogenetic Analyses

*Antiqusolidus maculatus* is placed incertae sedis, and according to the character comparison and phylogenetic analyses, *Antiqusolidus* is probably closely related to the subfamily Rutelinae. *Antiqusolidus* can be easily placed in the superfamily Scarabaeoidea due to the following key synapomorphies: antenna with club; prothorax highly modified for burrowing, with large, tubular coxae and dentate protibiae with only one spur; and tergite VIII forming a true pygidium and not concealed by the tergite VII [37,38]. Its assignment to the family Scarabaeidae is likely due to the following characters: eye divided by a canthus (excluding Ochodaeidae), venter smooth without dense setae (excluding Pleocomidae, Glaphyridae), six visible abdominal ventrites (excluding Lucanidae, Passalidae, Diphyllostomatidae, Trogidae, Glaresidae, Diphyllostomatidae), and pygidium exposed (excluding Geotrupidae, Hybosoridae, Belohinidae) [38].

The assignment of *Antiqusolidus* to any of the other subfamilies of Scarabaeidae can be reasonably excluded by the combination of the following characters: clypeus without teeth (excluding it from Chironinae), eyes large, visible from above (excluding it from Aegialinae, Eremazinae, Aulonocneminae, Termitotroginae), antennal insertions visible from above (excluding it from Dynamopodinae), scutellum exposed (excluding it from Scarabaeinae), pygidium exposed (excluding it from Aphodiinae), mesepimeron not protruding and invisible from above at the base of elytron (excluding it from Cetoniinae), metaventrite not longer than abdominal sternites (excluding it from Orphininae and Allidiostomatinae), outer edge of protibial with three teeth (excluding it from Phaenomeridinae), and protibial spurs present (excluding it from Aclopinae) [39,40,41,42]. From the remaining three subfamilies, namely, Melolonthinae, Dynastinae and Rutelinae, the new fossils show the following differences (Table 1) [37,43]. According to results based on the available characters, *Antiqusolidus* seems to be more similar to Rutelinae.

Our phylogenetic analysis supports that *Antiqusolidus* is closely related to Rutelinae. The MP analysis using the traditional search resulted in the two most parsimonious trees with 179 steps, a consistency index (CI) = 0.36 and a retention index (RI) = 0.71. MP analysis supported the placement of *Antiqusolidus* gen. n. in one of the phytophagous lineages of pleurostict Scarabaeidae (Appendix A). In the strict consensus tree, *Antiqusolidus* gen. n. was nested within the clade Rutelinae, which, however, was polytomous and not better resolved (Bremer support = 1). The current phylogenetic placement of *Antiqusolidus* gen. n. must, however, be considered preliminary, since scarab dung beetles were nested in this analysis within the pleurostict Scarabaeidae (Appendix A); their monophyly has been proven in many previous studies [16,44].

### 3.3. The Function of the Exaggerated Hind Legs

Here, we tested two hypotheses about the function of the extremely modified hind legs: “springing” and “fighting”. To test the hypothesis of a “springing” function, the studied 3D models were classified into two groups: the normal-sized spur group (Figure 3h,i, *Repsimus manicatus*, *Gastroserica kucerai*) and the exaggerated spur group (Figure 3f,g, *Aphthonoides armipes*, *Antiqusolidus maculatus* gen *et* sp. n.). The comparison of the maximal von Mises stress among all models (Figure 3e) shows that the values of the maximal von Mises stress (MMS) of the normal-sized spur group are significantly higher than those of the exaggerated spur group. Specifically, within the exaggerated spur group, the MMS of *Aphthonoides armipes* is lower than that of *Antiqusolidus maculatus* sp. n. The distributions of the von Mises stress in all models were greatly unbalanced. Most high von Mises stresses were focused or near the tips of spurs (Figure 3f–i). Based on the Concentration Index (CI) of the high von Mises stress (top 50% of maximum von Mises stresses) defined in this study, which reflected a mechanical failure, we found that the Concentration Index (CI) of the normal-sized spur group is significantly lower than that of species in the exaggerated spur group. In other words, the value of the Concentration Index (CI) is lower, and the high von Mises stress will be more focused on limited elements of the FEA model. The results indicated that exaggerated spurs could disperse more external force (lower MMS when under the same force); in return, they could generate a higher counter-acting pushing force when moving.

Another possible function of the exaggerated hind legs of the fossil taxon is their use as devices associated with fighting for access to females. The values for the two indices (FSI, TSI) of *Antiqusolidus maculatus* were within the range of the male monkey beetle with exaggerated hind legs (Figure 4, red star), which could suggest the fighting function of the hind legs of *Antiqusolidus maculatus*. *Repsimus manicatus* is a typical species from Rutelinae with exaggerated hind legs, which is also within the range (Figure 4, green circle). The other two species studied (*Aphthonoides armipes*, *Gastroserica kucerai*) are divided from the above two species with FSIs and TSIs closer to the ranges of non-fighting Hopliini (Figure 4, black triangle and orange square), which is consistent with their behavioral tendency to not use their hind legs to fight.

## 4. Discussion

### 4.1. Phylogenetic Position of Antiqusolidus gen. n.

Although the results of our phylogenetic analysis are not entirely consistent with currently accepted phylogenetic hypotheses for Scarabaeoidea [16,44,45], our findings provide some evidence that *Antiqusolidus* is closely related to the subfamily Rutelinae. Consequently, our results might shed new light on the timing of the origin of phytophagous scarab beetles in the Early Cretaceous. Our new taxon would represent one of the earliest records of Mesozoic pleurostict scarab beetles and can contribute to the study of their divergence time and of related lineages based on molecular evidence calibrated using the fossil record or using total-evidence phylogenetics. *Antiqusolidus* gen. n. supports that phytophagous scarab beetles arose at the beginning of angiosperm dominance [45,46] (orange star in Figure 5).

#### 4.1.1. The Earliest Record of Phytophagous Scarab Beetles

Normally, scarabaeoids feed on a wide range of plant and animal matter and can be basically separated based on two kinds of feeding habits: phytophagous and saprophagous [47]. Based on the fossil record, it was formerly widely believed that the phytophagous chafers first appeared in the Cenozoic after the diversification of angiosperms [48]. Recently, findings from some fossil evidence from the Mesozoic but also fossil-calibrated molecular phylogenies have rejected that hypothesis. To date, phytophagous scarab beetles are known from the Early Cretaceous, including Melolonthinae *Cretomelolontha transbaikalica* (the Early Cretaceous 125–112 Ma), some species belonging to Sericini (the Early Cretaceous ~125 Ma) and some glaphyrides (e.g., *Cretoglaphyrus* from the Early Cretaceous ~125–122 Ma) [46,49,50]. However, the systematic assignment of some of them is debatable due to their poor preservation (e.g., *Cretoglaphyrus* and Sericini). Based on comparative morphological evidence and the results of our phylogenetic analysis, *Antiqusolidus* gen. n. is possibly closely related to Rutelinae and thus to the generally phytophagous pleurostict scarab subfamilies (Melolonthinae, Dynastinae and Rutelinae) [16].

It is hard to determine what kinds of plants and which parts were eaten by *Antiqusolidus*. In the age of its occurrence, gymnosperms occupied most terrestrial habitats, whereas angiosperms appeared soon after (green line and red line in Figure 5). As a plant visitor, *Antiqusolidus* chafers might have been associated with both of them, but normally, beetles that are presumably closely related to *Antiqusolidus* prefer angiosperms because gymnosperms have hard leaves and special aromatic hydrocarbons, except for some beetles that were small enough to dig into the seeds or stems related to the gymnosperms, for example, the beetle families Scolytidae and Boganiidae [51,52]. Furthermore, the soil and litter revolution triggered by angiosperms provided a more suitable habitat for larvae of scarab beetles [16]. After comparing the size and structure of gymnosperms and angiosperms recorded in the same age, we think this Rutelinae beetle was likely a visitor of basal angiosperms or a visitor of gymnosperm lineages: Bennettitales (e.g., *Williamsonia*, Bennettitales) (Appendix A).

#### 4.1.2. Intraspecific Morphological Variability

We found some variations within *Antiqusolidus maculatus* regarding the length and position of the elongated metatibial process (Figure 1c,h). This variability could be interpreted as sexually dimorphic hind legs, which also appear in many extant insects, such as the subtribe Heterosternina of Rutelinae (e.g., *Heterosternus*, *Promacropoides* and *Macropoides*) [53,54,55,56], Hopliini of Melolonthinae (e.g., *Pachycnema*, *Heterochelus*, *Denticnema* and *Hoplocnemis*) [35], Macrodactylini of Melolonthinae [57], leaf beetles (*Sargra femorata*, Chrysomelidae) [58], leaf-footed bugs (Coreidae, Hemiptera) and several others [59,60]. Meanwhile, the characters of exaggerated spurs and color marking patterns of the body in holo- and paratypes could be considered stable characters of this species.

### 4.2. Exaggerated Hind Legs and Their Possible Functions

*Antiqusolidus maculatus* gen. *et* sp. n. has unusually robust and structured hind legs, consisting of a thick and strong metafemur and a metatibia armed with greatly enlarged spurs and an elongated process (Figure 1f,j). Normally, predation and feeding traits are associated with exaggerated structures in the anterior part of the body, such as forelegs (e.g., praying mantises), mouthparts (e.g., larval antlions, some weevils or soapberry bugs) and antennae (e.g., genus *Streblocera* in Braconidae) [61]. Traits associated with locomotion, e.g., jumping, swimming or digging, seem to be possibly related to exaggerated hind legs (e.g., flea beetles, water beetles or dung beetles) [62,63,64]. Furthermore, organisms with jumping legs are often recognizable by their enlarged femora and elongated tibia, such as in Orthoptera and flea beetles. Correspondingly, swimmers have coxae that are immovably fused to the thorax, a streamlined body, retractable hind legs or inflexible hairs in legs; diggers often have short femora, strong and toothed tibiae, and weak tarsi [13]. Clearly, the structures in this fossil are unique and special, which is not immediately related to any known typical exaggerated insect structures or their specific functions, as we have discussed before. In contrast, sexual selection seems to be a possible hypothesis as long as any exaggerated structures are different between males and females. After a close examination of the fossils, although their sex remains unknown to us, we find some clues that favor the two hypotheses about “springing” and/or “fighting”.

#### 4.2.1. Exaggerated Spurs on Hind Legs with Possible “Springing” Function

The tibiae in beetles commonly bear a pair of spurs on the inner margins [65]. It is strange that this fossil has unusually large metatibial spurs on the outer margin. The “springing” hypothesis would imply that *Antiqusolidus* gen. n. cannot make real jumps like Orthoptera but that the spurs can provide bumping protection when the beetle falls down and provide a higher pushing force when moving, which could be summarized as “springing”. This feature is known to exist in a genus of the flea beetle, *Aphthonoides,* which has a very long spur in the hind tibia that is as long as the tibia (Figure 3a,f). *Aphthonoides* is also believed to be a low-speed jumper [14]. In this fossil, the function of the developed femur, strong tibia and long spurs is probably related to small bouncing movements from the ground to lower plants (for example, a lower angiosperm from the Early Cretaceous: *Archaefructus sinensis*) or bumping protection when escaping from predators.

The “springing” hypothesis was supported by the FEA, which showed a lower MMS and lower concentration of von Mises stress in the fossil than the normal-sized spur group. Furthermore, the similar pattern of the von Mises stress of *Antiqusolidus* gen. n. and *Aphthonoides* provides further evidence for the similar spring function of the hind legs. The actual values of von Mises stresses and strains developed in this study should be interpreted with caution because of the complicated bio-structures and our simplified assumptions of homogeneous and isotropic material properties. However, we can interpret the relative performance of different legs with confidence to draw qualitative conclusions by applying identical material properties and scaling the models appropriately [66].

#### 4.2.2. Elongated Process with Possible “Fighting” Function

The alternative and second possible hypothesis of a “fighting” function would mean that the structures can be used as devices to fight with competitive males to enhance the successful passing of its genes [67]. Only limited cases of mating behavior have been reported in the Mesozoic fossil record, including respective structures such as the pronotum, antennae, abdominal segments, etc. [9,68,69,70,71]. *Antiqusolidus* gen. n. is probably the oldest record with hind leg dimorphism for beetles. Based on their unique shape and structure, and also considering our knowledge of the biology of extant scarab beetles, it is most plausible that these hind legs with their elongated process were used as a device to drive away competitive male rivals. To further understand their lifestyle, behavior and function, more evidence from this extinct group is needed.

### 4.3. Possible Biology Inferred from Marking Pattern

Preserved color markings are rare and generally lost during fossilization. Such records are so far limited to membranous wings, such as those of Orthoptera, Homoptera and Neuroptera, or parts of abdomens. Such preserved characteristics (e.g., delicate markings) are so far lacking in beetles [72,73,74]. Interestingly, color marking patterns almost cover the whole body and elytra of *A. maculatus* gen. *et* sp. n., which is especially obvious in the holotype.

In beetles, while diurnal and nocturnal species are both highly diverse in their color patterns, diurnal species generally have a higher frequency of being multicolored or having color patterns [75]. This generally goes hand in hand with mimetism or mimicry [76,77]. Focusing on scarab beetles, most extant chafers are simply and uniformly colored, being inconspicuous in their natural environment (nocturnal species often have colors from dull brown to yellow, and diurnal species have various shades of green or yellow that are in harmony with their surroundings) [78,79]. According to the data collected (Appendix A), shining leaf chafers with bright marking patterns (e.g., Parastasiina and *Rutela*) are diurnal and sometimes flower visitors [80,81]. Furthermore, marking patterns are well preserved for this fossil beetle and have great similarity to the extant Rutelinae (e.g., *Lutera nigrita*, *Promacropoides gloriagaitalis* and *Rutela dorcyi*) [53,80]. Besides Rutelinae, some similar marking patterns also existed in several diurnal beetles, such as Melolonthine and rose chafers [38,81]. Overall, the marking patterns of *Antiqusolidus* suggest that they were active plant feeders in the daytime and might have also been flower visitors [82].

Alternatively, the marking pattern of *Antiqusolidus* could in fact be mimicry. In this hypothesis, *Antiqusolidus maculatus* might have used its markings, especially the transverse belt markings on the abdomen, to imitate vespoid wasps (Vespoidea), which have stingers to give a warning to the predators. However, this assumption can be rejected since the origin of vespoid wasps is far later than when *Antiqusolidus* occurred [18].

## 5. Conclusions

The present study describes a new species belonging to the family Scarabaeidae, *Antiqusolidus maculatus* gen. *et* sp. n., from the Early Cretaceous. The new species has greatly enlarged spurs and an elongated metatibial process, which may have been used for springing movements, for bumping protection after purposely falling off, as a pushing force when walking to avoid predation, or for fighting with competitive males for potential mates. In addition, the unusually sophisticated marking patterns on the pronotum, elytra, pygidium, abdomen and legs suggest that the new species might have been diurnal and potentially visited the exposed parts of plants. Future discoveries of more well-preserved fossil beetles for in-depth studies are expected to further contribute to our knowledge of ancient behaviors.

## Figures and Tables

**Figure 1 biology-12-00237-f001:**
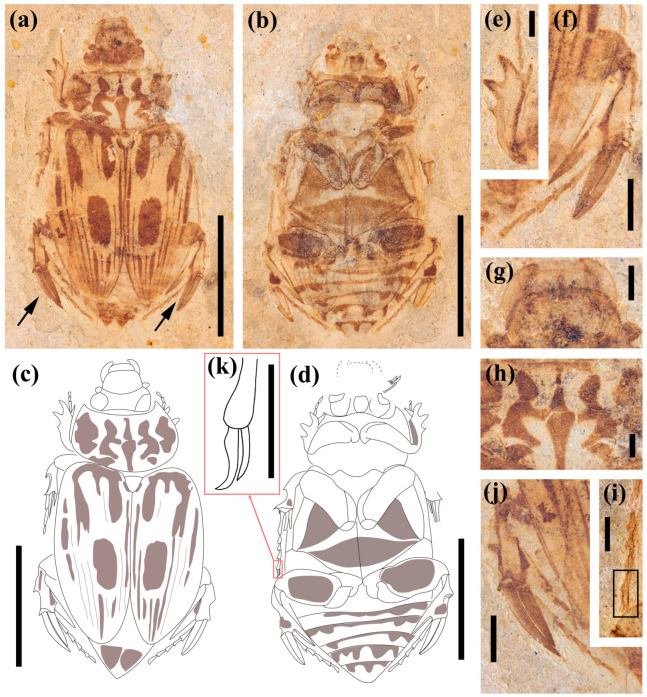
Photographs and line drawings of *Antiqusolidus maculatus* Lu, Ahrens, Bai, Shih & Ren gen. *et* sp.n. Holotype. (**a**,**c**) General habitus, dorsal view; arrows indicate structured hind legs, scale 10 mm. (**b**,**d**) General habitus, ventral view, scale 10 mm. (**e**) Left foreleg, dorsal view, scale 1 mm. (**f**) Right hind leg, dorsal view, scale 2 mm. (**g**) Anterior part of head, dorsal view, scale 1 mm. (**h**) Pronotum, scale 2 mm. (**i**) Right mesotarsi, ventral view, rectangle indicate claws, scale 1 mm. (**j**) Left hind leg, dorsal view, scale 2 mm. (**k**) Claws of right mesotarsi, scale 1 mm.

**Figure 2 biology-12-00237-f002:**
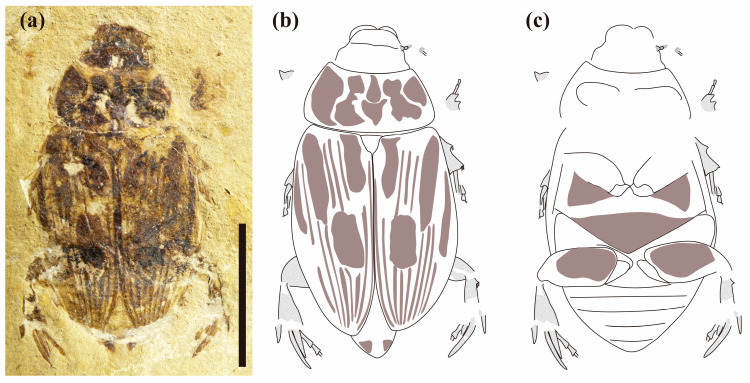
Photographs and line drawings of *Antiqusolidus maculatus* Lu, Ahrens, Bai, Shih & Ren gen. *et* sp.n. Paratype. (**a**) General habitus, scale 10 mm; (**b**) general habitus, dorsal view; (**c**) general habitus, ventral view.

**Figure 3 biology-12-00237-f003:**
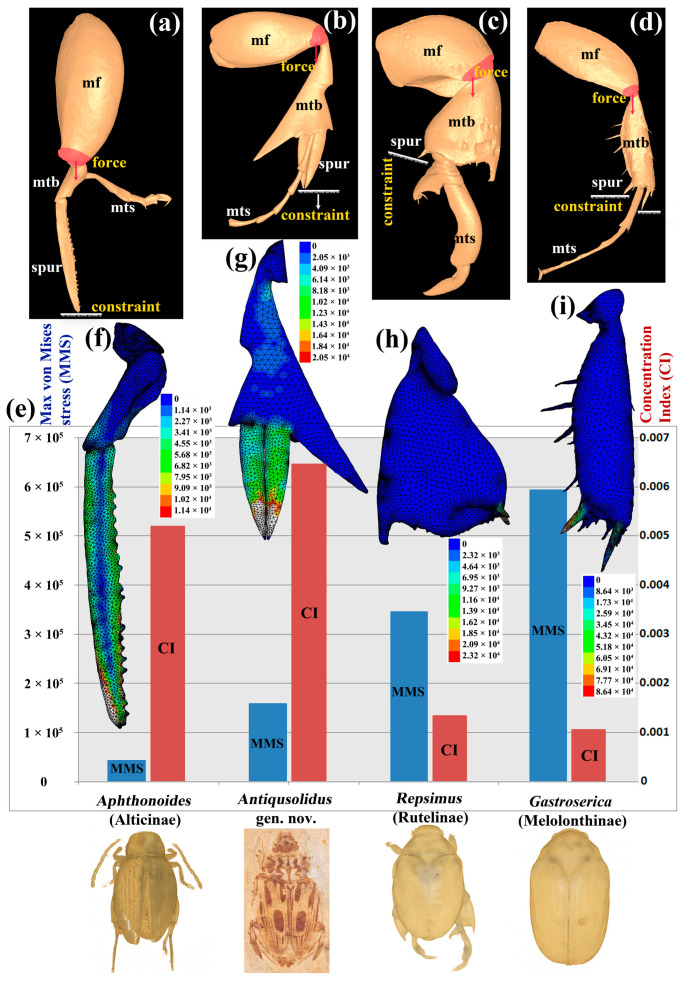
Possible springing function of the hind legs using FEA simulation. (**a**–**d**) Three-dimeniosnal models of hind legs. Boundary conditions: all models were transformed in Geomagic Studio 12 with the longer spur of hind tibiae in the Y-direction; the constraints were set at the tip of the spurs. The force loads were applied to the hind tibial base in the Y-direction; the boundary condition sets mimic the scene when the beetle falls down. Abbreviations: mf: metafemur; mtb: metatibia; mts: metatarsi; sp: spur. (**e**) The comparison of the maximal von Mises stress (MMS, blue histogram) and Concentration Index (CI, red histogram) of the high von Mises stress among all models. (**f**–**i**) Three-dimensional hind legs with von Mises stress contours. (**a**,**f**) *Aphthonoides armipes* (Alticinae). (**b**,**g**) *Antiqusolidus maculatus* sp. n. (fossil). (**c**,**h**) *Repsimus manicatus* (Rutelinae). (**d**,**i**) *Gastroserica kucerai* (Melolonthinae).

**Figure 4 biology-12-00237-f004:**
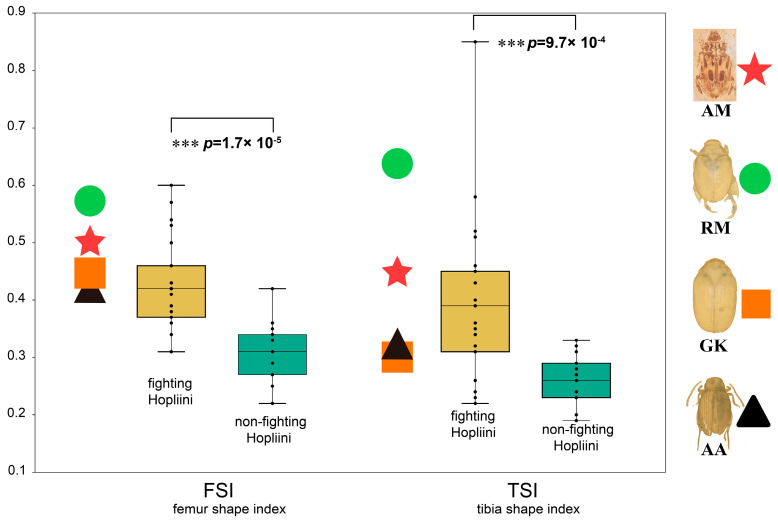
Fighting function of the hind legs based on the morphometric analyses of FSI and TSI compared between fighting and non-fighting Hopliini. Abbreviations: AA: *Aphthonoides armipes* (Alticinae); AM: *Antiqusolidus maculatus* sp. n. (fossil); FSI: femur shape index; GK: *Gastroserica kucerai* (Melolonthinae); TSI: tibia shape index; RM: *Repsimus manicatus* (Rutelinae). *** *p* < 0.001.

**Figure 5 biology-12-00237-f005:**
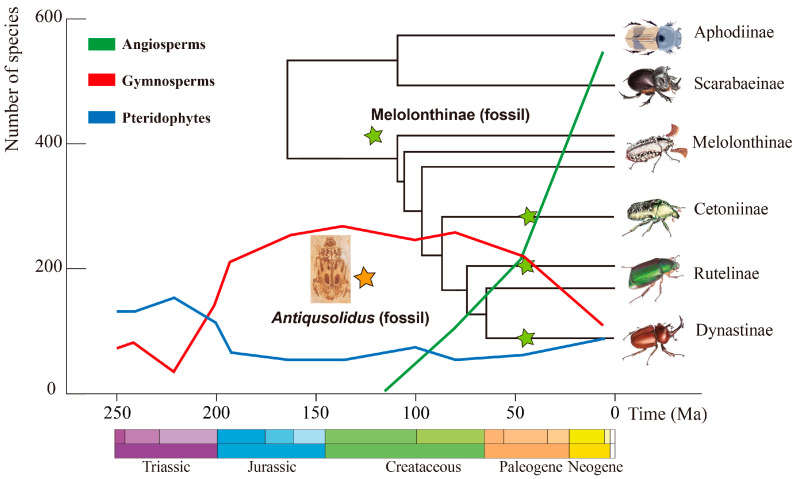
Diagram of the relationships between fossil record of Scarabaeidae and plant diversity in Phanerozoic. The phylogenetic tree and divergence time of some Scarabaeidae lineages were adapted from Ahrens et al. [16]. The three colored curves represent the diversity of plant species (blue line for pteridophytes, red line for gymnosperms, green line for angiosperms) throughout the Phanerozoic (adapted from 80). Stars represent the earliest fossil records in the related group: *Antiqusolidus maculatus* gen. *et* sp. n. (orange star, ~125 Ma) and *Pelidnotites atavus* Cockerell, 1920 (green stars, 48.6–40.4 Ma), for Rutelinae; *Cretomelolontha transbaikalica* Nikolajev, 1998 (125–112 Ma), for Melolonthinae; undescribed cetoniine (40.4–48.6 Ma) for Cetoniinae; and *Oryctoantiquus borealis* Ratcliffe & Smith, 2005 (44.6–46.8 Ma); for Dynastinae (green stars).

**Table 1 biology-12-00237-t001:** Key characters to distinguish Melolonthinae, Dynastinae, Rutelinae and *Antiqusolidus* gen.n.

Characters	Melolonthinae	Dynastinae	Rutelinae	*Antiqusolidus* gen.n.
Mandible	Invisible from the dorsal view	**Visible** from the dorsal view	Invisible from the dorsal view, or sometimes slightly visible	**Visible** from the dorsal view
Labrum	At least **partly visible**, or concealed beneath clypeus or apparently absent	Concealed beneath clypeus or apparently absent	At least **partly visible**, or concealed beneath clypeus or apparently absent	**Partly visible**
Color	Usually reddish brown or black, sometimes with metallic blue or green luster or distinctly marked with patches of scales	Usually testaceous, brown or black	Dull browns and yellows (nocturnal species) to **brightly patterned** and brilliantly metallic, even in silver and gold	**Color marking pattern**
Claws	Simple, cleft, toothed, serrate or pectinate usually paired, equal in thickness and length	All subequal in size	Unequal in length or size, and frequently weakly split at apex; one claw of each pair reduced	Seems unequal in length or size
Eyes	Not or only slightly protuberant, or **strongly protuberant**	Not or only slightly protuberant	Not or only slightly protuberant, or **strongly protuberant**	**Strongly protuberant**
Frontoclypeal suture	Absent or incomplete, or indistinctly impressed, or **distinctly impressed**	Absent or incomplete	Absent or incomplete, or indistinctly impressed, or **distinctly impressed**	**Distinctly impressed**
Ratio of elytral length to pronotum length	**1.55–4.55**	0.45–2.52	**1.6–4.1**	**3**
Mesoventral process	**Absent** or not extending to middle of mesocoxal cavity, or extending at least to middle of mesocoxal cavity	Extending at least to middle of mesocoxal cavity	**Absent** or not extending to middle of mesocoxal cavity, or extending at least to middle of mesocoxal cavity	**Absent**
Abdominal process	Broadly rounded or angulate, or absent	**Acute or narrowly rounded**, or broadly rounded or angulate	**Acute or narrowly rounded**, or broadly rounded or angulate, or absent	**Acute or narrowly rounded**
Tarsi	**Normally thin and undeveloped**	Normally stubby and developed	Normally stubby and developed	**Thin and undeveloped**

## Data Availability

The data presented in this study are available in the article. Further information and requests for the specimen should be directed to the Lead Contacts, Ming Bai (baim@ioz.ac.cn) and Dong Ren (rendong@mail.cnu.edu.cn).

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
