# Peer review of "A Cretaceous Chafer Beetle (Coleoptera: Scarabaeidae) with Exaggerated Hind Legs—Insight from Comparative Functional Morphology into a Possible Spring Movement"

_biology, 2023, doi:10.3390/biology12020237_

Round 1

Reviewer 1 Report

The manuscript looks very nicely and present very interesting and important finding of a new fossil scarab beetle. Authors made an effort to conduct several different analyzed to investigate taxonomic position of the new taxon but also function of the very peculiarly structured tibial spurs on hind legs. However, it is difficult to judge what was the purpose of these structures, as the environmental conditions (including plants), although narration of the authors is interesting. There are several other modern groups with exaggerated hind leg spurs including Melandryidae (Orchesia, Lederina), Scraptiidae, Bruchinae, and still don’t know the purpose of these structures (most probably not for fighting), but most of these beetles are phyto- or mycophagous. The MS is well written and I recommend its publication as it is.

Author Response

Dear reviewer:

Thank you for your positive opinion!  Yes, it is difficult to judge the function of extinct structures. In this work, we hope to explore the possible functions as much as we can and provide more clues about the evolution of insects.

Thanks again!

Reviewer 2 Report

This paper reports the earliest record of phytophagous scarab beetles. The new species shows well preserved marking patterns and unusually robust and structured hind legs with greatly enlarged spurs and elongated process. Based on adequate data acquired from their own specimens and references, functions of the exaggerated characters are analyzed and life habits of the extinct species are discussed. A few suggestions have been made and marked in the attached PDF. Besides, the author needs to pay more attention to the grammatical issues in the paper.

Author Response

Dear Reviewer:

Thank you for all your useful advices! We have revised this manuscript as follows:

1 English language have edited carefully through whole manuscript based on two people and one of them is British entomologist. The sentences edited by reviewer due to language all have been revised.

2 According to the reviewer’s opinion, the part of “Morphometric analyses for the fighting function of the hind legs” have be revised. The previous problematic analysis have been abandon. Please see the new analysis in the revised manuscript.

Thanks again for all your efforts!